# The Naked Truth: Temptation and the Likely 'Fall' of Catholic Education

**David Torevell and Michael James Bennett \***

Department of Theology, Philosophy and Religious Studies, Liverpool Hope University, Liverpool L16 9JD, UK; toreved@hope.ac.uk
\* Correspondence: mbennett@alldsaintschs.org.uk

**Abstract:** This article highlights one likely 'fall' to which Catholic education is susceptible in the modern era due to the oppressive climate in which it operates. Our critical method in arguing for this position is to oscillate between two texts—one written and one visual: *Genesis 3: 1–18* and Masaccio's painting of 'The Expulsion'. The hope is that one will inform and enrich a deeper understanding of the other. As part of this exercise in creative hermeneutics, we first argue that the dramatic story of the fall through pride or amor sui (self-love) and its resultant feeling of shame is a universal one in which readers (listeners) glimpse the long history of their own fears and desires. Second, we show how one 15th century Italian painter represented the tragic consequences of the Faustian self by examining Masaccio's painting in some detail. Third, we investigate St. Augustine's writings on this narrative and suggest how some forms of self-elevation align dangerously with the promotion of the autonomous self in contemporary education. We also critically examine exegetical writings from Jewish and Christian perspectives to draw out further meanings of the narrative. Fourth, we point to the themes of hiding and forgiveness embedded in the account which leads us neatly into the last fifth section where we discuss the text's implications for contemporary Catholic education. Here, the focus is on one likely 'fall' of Catholic education when it fails to live up to its distinctive mission to place love unconditionally at its centre. In a highly market-driven, managerial climate of competition where league tables, bureaucratisation, and data analysis assume an overwhelming significance allied to institutional survival and kudos, the temptation is to show the worth of the school by emphasising its examination success and employment rates rather than through its service to others, especially those who have been forgotten. Although we are highly sensitive to the conflictual demands on Catholic institutions at the present time from a variety of stakeholders, we conclude that their healthy continuation depends on their public, ethical avowal to love everyone unreservedly with assistance from God's grace and when this aspiration fails, to seek forgiveness. The article is not concerned with strategies of resistance against those developments in education contrary to a Catholic philosophy.

**Keywords:** Genesis; fall; Catholic; Catholic education

## 1. Introduction

This article highlights the likely 'fall' to which Catholic education is susceptible in the modern era. In arguing for this position, our biblical hermeneutical method oscillates between two seminal texts about people's temptation to elevate themselves too highly. One is written *Genesis 3: 1–18* and one is visual Macaccio's *The Expulsion* (1425–1427). We suggest that this narrative is particularly appealing because it deals honestly with key debilitating aspects of the human condition—pride, loss, shame, and guilt—but, at the same time, offers a way out of these entrapments through the humility to ask for forgiveness. The American Jewish literary critic Greenblatt argues that the story of Adam and Eve continues to enrapture treaders and listeners from across and beyond the boundaries of religious traditions due to its mirroring nature: 'For reasons that are at once tantalising and elusive, the few verses of the story of Adam and Eve in an ancient book have served as a mirror in

which we seem to glimpse the whole, long history of our fears and desires. It has been both liberating and destructive, a hymn to human responsibility and a dark fable about human wretchedness, a celebration of daring and an incitement to violent misogyny' (Greenblatt 2018, pp. 5–6)[1]. This article endorses this position and is a dominant strand throughout our argument.

Our article is presented in five sections:

A universal story of envy, pride, shame, and loss

Masaccio's The Expulsion

St Augustine's insights into the text versus the modern educational turn towards the autonomous self

Shame, hiding, and the possibility of forgiveness

The 'fall' Of Catholic education

## 2. A Universal Story of Envy, Pride, Shame, and Loss

The Jewish and Christian seminal account of Adam and Eve's sin in *Genesis 3: 1–18* is essentially an anthropological narrative concerning who human beings are and to what they are dangerously susceptible. As a universal text with a timeless significance, it appeals to all humanity, not simply Jews and Christians. Critical of (post)modern writers who suggest such universalism is ill-advised, we unashamedly disagree with this contention. Along with Eagleton, we insist that religious thinking and texts posit a common humanity easily identifiable. As he comments, 'a God who concerned himself only with a particular section of the species, say Bosnians or people over five foot eight inches tall, would appear lacking in the impartial benevolence appropriate to a Supreme Being' (Eagleton 2015, p. 188). Biblical texts have the advantage of being timeless and pertinent to all cultures and histories.

The dramatic story has become traditionally known as the 'fall' of humanity and involves five characters—God, Adam, Eve, the devil, and the cherubim—who become embroiled in a personal and cosmic battle. Much Jewish and Christian life and theology rest upon this spiritual warfare, the consequences of which can be either redemptive or devastating. The narrative focusses on the importance of obeying God's commands and staying in a loving relationship with Him and exhorts the living of a virtuous life through the following of God's will, not our own. The dangers of pride, envy, self-aggrandisement, and rivalry are signalled throughout the story, as it explores these human traits in a highly dramatic manner.

The word 'Adam' comes from the Hebrew noun '*ha Adamah*' which means 'earth'. Readers are remined how 'the Lord God formed man of dust from the earth' (*Gen 2, 7*) a theological truth about humanity's mortality and creaturehood. Unequal to God, humanity's challenge is not to be in rivalry with Him, with others or with ourselves. The author of *Ecclesiasticus* echoes a similar warning: 'Do not exalt yourself, or you may fall and bring dishonour upon yourself' (*1, 30*). Atheists, agnostics, and religious people alike surely recognise this commonly felt temptation.

As testimonies to its enduring significance, the episode has lent itself to multiple creative mythologisations in the creative and performing arts. One of the greatest of all English epic poems is directly based on the narrative, Milton's (1608–1674) *Paradise Lost*. Its opening lines reveal starkly the consequences of not obeying God's commands: *'Of man's first disobedience, and the fruit/Of that forbidden tree, whose mortal taste/Brought death into the world . . . '* (opening verse) and Book Nine records the quandary Adam finds himself in—if he eats the apple, he will lose paradise; if he does not, he loses Eve, a theme we shall return to later in our exposition of Masaccio's fresco. Marlowe's (1564–1593) renowned play *Dr Faustus* addresses the same issue by staging how the protagonist attempts to usurp divine power and knowledge and the distressing consequences of such hubris. The chorus tells the audience at the beginning of the play how the protagonist's *'waxen wings did mount above his reach/And melting heavens conspired his overthrow'* (Marlowe 1972, p. 265). There is an allusion to the Greek myth of Icarus here, who wishing to leave Crete, tried to fly, but soon

realised he was a mere human being. His 'waxen wings' melted when he drew near the sun, making him plunge to his death. He literally and metaphorically ascended too much for his own good. Peter Paul Rubens' (1577–1640) *The Fall of Icarus* (1636, *Musees Royaux Des Beaux-Arts*, Brussels) shows Daedalus, the father of Icarus, watching his son's tragic fall to his death, an incident Ovid (b. 43 BCE) writes about in *Metamorphosis.* Both Faustus and Icarus have 'swollen heads'; both thought too highly of themselves. In personal terms, the narrative warns against the dangers of a misplaced elevation of the self.

Arguably, the greatest literary text which deals with the same theme is Shakespeare's *Othello*. Iago decides 'to abuse Othello's aer' (Act 1, 3, line 371) like the serpent in *Genesis*, penetrates the core of Othello's being, an inner space 'where I have garnered up my heart/Where either I must live or bear no life'. (Act 4, 2, line 56–57). Like Adam and Eve, he falls through pride at the erroneous suggestion that his wife is unfaithful. Both texts are also stories about shame, a trope revealed in St Peter's denial of his Master three times but 'when the cock crowed, he realised what he had done, went outside and wept bitterly' (*Luke 22, 62*). According to Augustine, Peter's character, was 'more wholesome when he wept than when he was pleased with himself and presumptuous' (610), as was Othello's when he wept as he confronted his wife with his false accusation of adultery in the bedchamber minutes before he murdered her: 'I must weep' (Act 4, 2, line 41; Act 5, 2, line 20). Towards the end of the play, we hear Othello, crippled by shame and self-hatred, condemn his sin and offer himself up for punishment: –'O cursed, cursed slave! Whip me, ye devils/From the possession of this heavenly sight! Blow me about in winds! Roast me in sulphur! Wash me in steep-down gulfs of liquid fire!/O Desdemon! Dead Desdemon! Dead! O! O!' (Act 5, 2, line 275–277). Here the audience witness a repentant sinner aware of his crime and his need to be punished. But his anguish at his own failings overrides any redemptive move towards accepting God's forgiveness allied to his own repentance. Consequently, like Adam and Eve, he finds himself in unremitting despair and kills himself. Iago remains devoid of such shame or repentance throughout the play.

Nakedness, Banishment and Toil

The word 'naked' (*arom*) is mentioned four times between *2, 25–3, 11*.

2, 25. 'And the man and his wife were naked, and were not ashamed',

3, 7. 'Then the eyes of both were opened and they knew that they were naked.'

3, 10. Adam—' . . . and I was afraid because I was naked; and I hid myself.'

3, 11. God—'Who told you that you were naked?'

Why is there this emphasis on nakedness? One possible answer revolves around the association between nakedness and truth in the Christian tradition. Christians are invariably encouraged to be naked and truthful before God; in other words, to be frankly honest. For the French philosopher and mystic Simone Weil (1909–1943), truth is not revealed except in nakedness and the expression the 'naked truth' is used commonly in much discourse. Revelation is frequently described as unveiling the truth about those things which were previously hidden or concealed. Nakedness has a clear association shame, the genitals, and sexual desire. The Greek word *aidoia* means genitals and is a derivative of the word *aidos* which means shame and similar terms are found in other languages. Aidos is the Greek goddess of shame and modesty. Although the *Genesis* text does not explicitly state that Adam and Eve's fault was of a sexual nature, the forbidden tree is symbolic of temptation and sensual desire. After eating the forbidden fruit, Adam and Eve feel shame and guilt and become scared as they hear 'the sound of the Lord God' walking in the garden (v.8) panicking that they might be seen by Him, which is the reason why they hid. Williams argues that the common reaction to shame 'is to cover oneself or to hide, and people naturally took steps to avoid the situations which called for it' (Williams 2008, p. 78). Those in the 17th century who covered over Masaccio's painting of Adam and Eve's nakedness with fig leaves due to the scandal it might cause exhibited this tendency to associate nakedness with shame and sexuality. The fig leaves were removed in the 1980s and the painting is now as Masaccio originally painted it. We shall have more to say about shame a little later.

Has God banished Adam and Eve or have they banished themselves, we might question? Whichever view we take, Masaccio certainly paints verses *23–24* with dramatic intensity. It is only with the coming of Christ that a realignment becomes possible. Masaccio uses a vast amount of space and shadows on the fresco to represent the rough ground on which Adam and Eve now stand and this reflects *verse 3*, *17*: 'Cursed is the ground because of you; in toil you shall eat of it all the days of your life; thorns and thistles it shall bring forth to you'. Richardson comments that the expulsion is 'no arbitrary divine decree, but the fact that wo(man) is unfit to live in paradise. They have made God's good world a place of strife and *rivalry* . . . and the road of redemption is long and steep' (Richardson 1959, p. 78).

### 3. Masaccio's *The Expulsion*

*The Expulsion* is based on *Genesis 3: 23–24*: ' . . . therefore, the Lord God sent him forth from the garden of Eden, to till the ground from which he was taken. He drove out the man; and at the east end of the garden of Eden he placed the cherubim, and a flaming sword which turned every way, to guard the way to the tree of life'. It was painted sometime between 1425 and 1427 in the Brancacci chapel in the Church of Santa Maria del Carmine in Florence, which has become known as the Sistine chapel of the early Renaissance. The fresco sits alongside his other work, *Tribute Money.* Masolini's (1383–1447) *The Temptation of Adam and Eve* is opposite but as Cole Ahl comments, there is a striking difference in the two artists' portrayals of the couple: 'The courtly elegance of Masolini's Adam and Eve contrasts paradigmatically to Masaccio's huddled, weeping figures. Their soft pearly flesh serves as a foil to the vigorously modelled, sculptural anatomy of Masaccio's progenitors, a visual encapsulation of their prelapsarian indolence on the one hand and their toil after their expulsion on the other' (Cole Ahl 2002, p. 154). Vasari, the sixteenth century biographer of Masaccio suggested that he freed himself from Giotto's style and brought in a 'modern style' which began to be copied numerous times (Ames-Lewis 2002, p. 207).

Masolini's and Masaccio's other paintings cover the rest of the walls, all of which represent episodes in the life of St. Peter. Masaccio was the student of Masolini and 21 years his younger. Born in a small town just outside Florence in 1401, his real name was Tommaso di Ser Giovanni di Simone Cassai. He became known as Masaccio which means 'clumsy Tom' because he took no interest in his appearance, people, or politics due to his absolute focus on art. Two major influences, aside from Giotto, were the architect Brunelleschi from whom he learnt linear perspective and the sculptor Donatello, whom he imitated in paint with his 'sculptural' depictions of figures like Adam. He died in Rome at the age of 26 in 1427, some say by poison administered by a rival artist, some say by the plague.

Masaccio's *Expulsion* is the first one a visitor sees on the left, high up, as s/he enters the chapel. What is the expulsion doing here? Does it suggest that the Church led by St. Peter offers the grace required to counter the Fall? Perhaps. It is painted on what is known as *buon* plaster and Masaccio would only have been able to paint small sections each day, otherwise the plaster would have been too dry. There is a sense of forced, strained movement echoing the verse 'therefore the Lord God *sent him forth* from the garden of Eden . . . He *drove out* the man . . . ' (*3, 23–24*). They appear to be pushed out of the garden; even the rays of light (now dark) seem to steer them away, just as much as the angel. The painting conveys a trajectory of *culpable* expulsion from the left—innocence and joy in the garden—to the right—toil and pain on the earth outside.

Chakrabarti writes that the artwork has a power over our hearing as well as our seeing and after nearly fifteen years of first looking at the fresco, he thinks 'it is the loudest painting I've ever encountered. I saw it while walking silently around Florence's Brancacci chapel. I was transfixed by this particular fresco. Nearly 15 years later, I can still hear that endless howl of despair as Adam and Eve are ordered out of the Garden of Eden. They appear to know that it is a crime they are no longer innocent. They look towards an off-stage future, out in the wilderness, as they leave everything they knew behind. On Eve's face, amid the bleak, downturned eyebrows and an open, keening mouth, Masaccio

perfectly captured the desolation of new grief. I was shocked to suddenly recognize myself in that expression' (https://www.artsy.net/article/artsy-editorial-artwork-changed-life-masaccios-expulsion-garden-eden, (accessed on 13 October 2021)).

The heavy, vertical line of the gate, as well as the menacing sword of the angel, seem to prevent re-entry. They are truly banished. The sword and rays of light (now black) to the left of the painting, aside from giving a feeling of rushing movement and a directional pushing out of Adam and Eve, would have been originally silver, representing the glorious world from which Adam and Eve had been expelled. The bright red and folds of the angel's garments and the carpet on which he lands, are in stark contrast to the darkness of the earth. The sword acts as an incentive to banishment and as a barrier to re-entry. The seventeenth century devotional writer St. Francis de Sales in *Treatise on the Love of God* interpreted this verse about the sword as the need for everyone to be pierced by divine love in order to live the Christian life. For him, the sword was a symbol of redemptive pain and Masaccio's fresco for some viewers will carry this hopeful message. See (De Sales 2015).

The angel's body is foreshortened which allows Masaccio to compress the body to allow the eye to think it is looking at the person from a distance. Such compression creates a sense of depth or three-dimensionality and of perspective. The artist wishes to indicate that this destiny-changing incident is taking place in a definitive space and time and communicates feelings of intense loss. Like a frozen frieze from a film, the artist captures the precise moment of guilt-ridden exposure and expulsion. The architectural features of the gates and the emotional impact were most likely influenced by Giotto's work. Masaccio creates a stunning visual effect, a colourful and dramatic scene of displacement long before computer-generated imagery.

Eve is modelled on Venus, the Roman goddess of love and sex, the Greek equivalent to Aphrodite. Like her, she covers in modesty her breasts and genitalia. She has a *contrapposto* posture—the Italian word for a human figure standing with most weight on one foot and the other relaxed, which causes the hips and shoulders to rest at opposite angles. The most famous examples of this are Michelangelo's *David* and Botticelli's *The Birth of Venus.* The stance gives the impression that the character is ready to move on quickly when necessary. Adam and Eve seem as if they are in transit to another location, which is clearly what Masaccio was attempting to do by his use of this technique.

The contrast between the dark, barren landscape outside the garden and the glorious radiant red of the angel's attire and carpet on which he lands conveys what they have lost. Here, the two worlds are at odds with each other. However, as Chakrabarti comments, although Adam and Eve find themselves in the midst of darkness, there is also a feeling of redemption, a sense that they are 'walking out to the light ... there is hope for them' (Chakrabarti 2015). Masaccio paints them moving towards some glimmer of brightness and post-incarnation viewers might interpret this hopeful depiction with New Testament eyes, assuaging any feeling that the artwork is definitely depressing. Nevertheless, Masaccio wants to display the seriousness of humanity's fall, not their future redemption. This echoes the Pauline theology of just how generous Christ's salvific work is in light of humanity's rejection of God's love. As he writes in his *First Letter to the Corinthians 15*, *45–48:* 'The first man Adam became a living being; the last Adam became a life-giving spirit ... The first man was from the earth, a man of dust, the second man is from heaven'. And *Romans 5*, *15:* 'But the free gift is not like the trespass. For if many died through one man's trespass, much more have the grace of God and the free gift in the grace of that man Jesus Christ abounded for many.' As Jesus Christ becomes the New Adam and Mary the New Eve, things are reversed, so much so that the Easter liturgy can proclaim the fall as a 'happy fault'. The *exsultet* is sung on Holy Saturday night with the words, '*O felix culpa quae talem et tantum meruit habere redemptorem*' ('O happy fault that gained for us so great, so glorious a redeemer').

The depiction of the bodies and gestures of the Edenic couple highlight their nakedness and by doing this Masaccio contradicts the earlier *verse 22* that 'the Lord God made for

Adam and his wife garments of skins and clothed them'. He wants to portray their exposure and shame by this emphasis. Adam's body is muscular and tensed, due to his anxious state of mind and his inner turmoil. The covering of his face suggests his psychological torment, whereas Eve displays the agonizing despair of her face. Light shines on both bodies from the right, to expose their guilt and shame. The vanishing point is situated on their torsos which focusses onlookers' eyes on their naked bodies. Greenblatt writes:

> Masaccio's unforgettable figures depend . . . on their overwhelming sense of embodiment . . . Adam's right foot still touches the threshold of Paradise, but not for long. They are in the world now, and unlike the angel who possesses wings, a beautiful garment, a sword, and a kind of magic carpet, the humans are utterly unprepared. . . . They are entering a very harsh environment, and they have nothing whatsoever to shield or protect them. From this perspective, Adam's penis, strikingly central in the fresco's composition once the overpainted fig leaves were removed, is less a sign his virility, than of his being what Shakespeare calls "unaccommodated man". . . . His Adam and Eve were no longer abstract, decorative emblems of human guilt; they were particular suffering people, who had bodies with volume, weight, and, above all, movement (Greenblatt 2018, p. 150).

Clifton (1999) makes the illuminating point that in the 15th century, there were deeply entrenched attitudes towards gender difference, at odds with arguments about gender fluidity which rage today. Masaccio would have been aware of these and he exploits them. Is he attempting to suggest ironically, as many do now, that sharp distinctions between genders are no longer helpful or valid? Although Eve's sin that led to the Fall is not actually specified as a lapse of chastity per se in the text, what has developed in Western Christianity is an association of sin with sexuality, and this owes much to the scriptural portrayal of the primeval couple. What developed over time was a relentless blaming of Eve for her role in tempting Adam. Masaccio's image reflects the association of shame and nakedness (and, by extension, sexuality), in the figure of Eve only, providing for Adam a different gesture which also signifies shame, but of a different kind. By portraying Eve's gestures of covering her breasts and genitalia, Masaccio identifies her sexuality as the location for her sin and shame (Clifton 1999, pp. 642–55).

It was widely recognized at the time that women's bodies were the sites of provocative and alluring temptation and that females were innately unable to control their emotions and lustful bodies. In contrast, Adam covers his face, not his body, and since his visage was associated with the mind at this time, Masaccio is suggesting that Adam loses his mind during the Fall and a corresponding contemplation of God. It has very little to do with the control of his body. He also loses his 'manliness' due to the 'unreasonable' behaviour of giving in to Eve's suggestion, since 15th century thinking associated maleness with reason. Clifton points to the late 14th century handbook for artists *Il libro dell' arte* by Cinnino Cennini which discusses human proportions. The work claims that males possess proportions but females do not and are therefore like irrational animals (Clifton 1999, p. 646). Is it any wonder Adam feels bad about himself—a man losing his manliness, what more shameful thing is there to endure, then or now? Men in the 15th century and beyond were also discouraged from public displays of emotion in keeping with their presumed rationality. Thus, Adam hides his face, lest he demonstrates 'unmanly' emotion. Eve, on the other hand, wails openly and publicly, her gesture of shame centred on her body and her provocative sexuality. Might it be farfetched to suggest that since Masaccio was so aware of the nonsense of entrenched gender differences and stereotypes that he painted them ironically to highlight their absurdity? Was Masaccio the first male feminist and the first in history to be aware of the reality of gender fluidity?

## 4. Augustine's versus (Post) Modernity's Turn towards the Autonomous Self

Augustine (354–430) writes that since there was no prior evil in the garden, the door of *pride* let it in. Adam and Eve succumbed to the belief that they 'will be like God' (*Gen*, *3*,

*5*), if they eat of the 'fruit of the tree' (*Gen*, *3*, *3*). They had forgotten that they were 'dust and unto dust thou shalt return' (*Gen*, *3.19*). They forsook the foundation on which their minds and hearts ought to rest—God and opted instead for their *own* foundation, blatantly raising themselves above the Divine. Augustine defines the nature of their pride 'And what is pride but an appetite for a perverse kind of elevation?' (Augustine 1998, p. 608). Enjoyment and rest consequently became turmoil. Contemporary Trappist monk Dom Eric Varden describes the incident in these terms:

> "By acting as he did, . . . Adam preferred his (own) criteria to those of his Maker. He, who, at first, had stood face to face with the flaming countenance of God, whose being reflected God's glory, yielded to presumption. He thought he subsisted at God's level by some quality intrinsic to himself. The tempter's trick was to speak of God as being jealous of man . . . If the ruse worked, it was because Adam, at some level, did consider himself God's equal." (Varden 2018, pp. 15–16)

Adam refused to forsake his life's companion Eve and instead became 'her companion in sin' (608). Augustine insists, ' . . . even though the woman committed the transgression because of the serpent's persuasion, and the man because of the woman's offer, the transgression was *nevertheless their own act*' (611). However, the falling away from their true nature as children of God did not entail for Adam and Eve a complete loss of their being, but it did fracture a part of it and like Narcissus himself, their personhood became diminished and lost strength: 'By striving after more, (wo)man is diminished; when he takes delight in his own self-sufficiency, he falls away from the One who truly suffices him' (610).

Augustine became obsessed with the book of Genesis throughout his life and spent fifteen years writing *The Literal Meaning of Genesis* and dealt with it again in *The City of God*. The story of Adam and Eve became for him a way of understanding his own and others' struggles with will and desire. It became an incessant problem for him throughout his life, enhanced by the guilt he felt about his 13-year relationship with his lover before his conversion. The toil each person endures outside the garden partly consists in the battle between unruly desire and will and it is a lifetime struggle. This is the curse. Augustine refers to 'voluptuous thoughts' he often had in his monastic cell towards the end of his life. Masaccio captures with intensity this sexualised sense of shame and of toil.

The enduring question Augustine asked himself was this: why is it impossible for the will to control desire? His answer was that this spiritual battle only came about after the fall. Desire was not in opposition to the will in the garden. As a neo-Platonist, Augustine believed that it would have been perfectly possible for Adam and Eve to have had 'unlustful' sexual intercourse, had they not fallen. However, they sinned before this took place. They would not have had the activity of turbulent lust in their flesh, but only the movement of peaceful will by which they commanded the other members of the body. He writes, 'How happy, then, were the first human beings, neither troubled by any disturbance of mind nor pained by any disorder of the body!' (603). They incurred the penalty of exile from paradise before they could unite in the task of propagation as a deliberate act undisturbed by passion. Thus, Adam and Eve's disobeying of God's command was all the greater in proportion to the ease with which it could have been avoided, precisely because their desire was not in opposition to their will. Nevertheless, they had free will. The conflict between selfish, sexual desire, and the will only came about *after* the Fall: 'For, now, the flesh is in such a condition that it simply cannot serve our will' (613) and it was 'right to be greatly ashamed of this lust, and it is right that the members which it moves or fails to move by its own right, so to speak, and not completely in accord with our will, should be called shameful ...' (615). He had his answer to his questionings. Or so he thought.

After the Fall, everything changed. Adam and Eve saw for the first time what they had never seen or felt before—that they were naked—and it filled them with shame and impelled them to reach for fig leaves to cover as a veil for when 'grace was removed and a punishment commensurate with their disobedience inflicted on them, there appeared a certain shameless movement of the body ...' (615). They became victims of both involuntary

desire and voluntary arousal for satisfaction. Up until this Fall, the Edenic couple had lived in perfect freedom. Now outside the garden, they lived imprisoned by their proclivity to succumb to desire, lust, and the wiles of the flesh. This is not only the Edenic couple's proclivity but every person's, since he believed this tendency was passed on through later generations. All forms of addiction—sexual, alcoholic, gambling, drug—are susceptible to uncontrollable desire which the will seems inadequate to resist. Desire features strongly in the Genesis account and the tree becomes the symbolic, sensual allure which 'was a delight to the eyes' and to 'be desired' (*Gen.3*, *6*).

Augustine's view of the devil is insightful. The proud fallen angel, envious by reason of that same pride which had induced him to turn away from God, encouraged Adam to gloat over subjects of his own, rather than be subject to God Himself. The devil became envious of Adam and Eve because they did not possess the unfallen nature which he possessed.[2] The method of the devil's attack on the Edenic couple is worth noting. Augustine says the devil transfigured as a serpent, chose to 'speak through this creature, slippery and moving in twisted coils' and sought to 'insinuate himself, by crafty suggestions, into the heart of man, whose unfallen state he envied now that he himself had fallen' (Augustine 1998, p. 606). This echoes *Psalm 12*, *2* where one translation reads how enemies tell lies which 'slide off their oily lips'. The serpent directly appeals to the vanity of Adam and Eve insinuating doubts about the virtuous nature of God Himself (*Gen 3*, *5*). Ephrem the Syrian (306–73) asks whether Adam and Eve became trapped in the imagined god-likeness that the serpent falsely provided?' and St Anselm adds that the self-elevation witnessed in the garden is centred around the devil not only wanting to be equal to God, but also to be greater than God by desiring what God did not want him to, because he put his own will above God's. For the modern the biblical commentator, Richardson, the serpent should be seen not as something external to our nature, but as a personification of human temptation and that disobedience and doubting are two parallel processes, quoting the Danish theologian Kierkegaard, 'It is hard to believe because it is hard to obey' to make his point (72).

Augustine has a warning about *amor sui* (self-love). Paradoxically, he teaches that it is humility which elevates the mind, for it ascends by making itself subject to God. Exaltation to God abases the mind, unlike the devil's exaltation of himself, which held a debilitating sway over him. In *City of God*, he advises his readers that love is the central dynamic when it comes the practice of Christianity. Existential problems which might arise throughout life can never be associated with love itself, only in relation to what chosen object love is directed toward. The right object of love is God and humanity—one's neighbour, made in the image of God. Other things are mere idols. He shows how 'in the one city of God, love of God has been given pride of place, and, in the other, love of self (*amor sui*)' (Augustine 1998, p. 609; O'Donovan 2005). Adam and Eve had already become too pleased with themselves before the devil's temptation and that is why Adam was susceptible to the claim that they will become like God. It connected to a mindset to which he was already moving. Adam and Eve would have been better aligned to becoming gods (since they were made in God's image) if they had 'clung to the highest and true ground of their being and not, in their pride, made themselves their own ground' (610). By proudly striving for more than what had been donated, they became diminished. Augustine uses the metaphor of light to describe this descent into self-love—Adam is delighted with himself as if he were his own light and therefore turned away from that Light 'which, if only he had been pleased with It instead, would have made the man himself a light' (610). St. Paul in his *Letter to the Philippians* previously denounced this latter tendency: 'Their god is the *belly*; and their glory is in their shame; their minds are set on earthly things' (*3*, *19*).

## 5. The Way of Redemption: Shame, Hiding, and the Offer of Forgiveness

The text records how Adam and Eve, on realising their guilt, 'knew that they were naked; and they sewed fig leaves together and made themselves aprons' (*Gen 3*, *7*). They were afraid that God would see them naked and ashamed so 'the man and his wife hid

themselves from the presence of the Lord God among the trees of the garden' (*Gen*, *3*, *8*). Adam replies to God's question 'Where are you?' with the words, 'I was afraid because I was naked; and I hid myself' (*Gen*, *3*, *10*), but God's reply to Adam is 'Who told you that you were naked?' (*Gen 3*, *11*). Presumably, before they sinned, they were not ashamed or fearful of their nakedness—like young, innocent children, they were blissfully unaware that they were. Innocence has nothing to do with cover up or hiding. However, once Adam and Eve realise they were naked and exposed after they felt guilt and shame, they attempt to blame others to avoid any further exposure of the truth. Adam says, accusing both God and his companion, 'The woman who thou gavest to be with me, she gave me the fruit of the tree . . . ' (*Gen*, *3*, *12*), and Eve also passed the buck when she said, 'The serpent beguiled me and I ate' (*Gen 3*, *13*). Never do they admit publicly that it was their *own* fault, nor do they take any responsibility for their action. Nevertheless, they feel ashamed of their behaviour. Masaccio captures this clamour of pain and shame with such intensity which for us outstrips the Norwegian expressionist Edvard Munch's (1909–1943) agonised painting of *The Scream.*

The Jewish author Zornberg, drawing from *midrashic* sources, comments that after the fall, Adam and Eve's existence shrinks and they lose all stability as they move out of the garden. Adam now has 'a spinelessness, a vapidity. A splendid being decomposes before our eyes' (Zornberg 1995, p. 24). Mascccio has represented this collapse by Adam's downward gaze, his face hidden and buried in his hands. Not looking up, not even a side glance to the one he loves. Eve's posture is different, but equally tragic as she covers her breast and genitals with her hands in shame, lest she be fully exposed. Rarely in Western art is the feeling of regret and shame so powerfully represented as in this fresco, and to some degree at least, it is a universal feeling. Don't we all know how they both felt?

Shame also often results, says Augustine, in isolation. In *de Trinitate*, he comments that the soul slipped away from the shared whole to its own restricted part. The two figures no longer have any eye contact between them and, what is worse, they have to endure this loneliness on a brutalised Earth, or as the narrative puts it, ' . . . thorns and thistles it shall bring forth' (*Gen*, *3*, *17–18*). Adam's and Eve's guilt and shame could have been easily reversed. God appears to them and offers his gentle voice of welcoming return 'at the cool of the day' (*Gen 3*, *8*). Symeon (949–1022) advises us that God searches for the Edenic couple because he loves them and God was not angered, nor did he immediately turn away. But he also adds on God's behalf, 'Do you think you can hide from me?' God's question to Eve, 'What is this you have done? '(*3*, *9*) gives her the opportunity to admit their fault. As the Orthodox Lenten Vespers states, 'Then the Saviour said to him: "I do not desire the loss of the creature I formed, but that he should be saved and come to the knowledge of the truth. For he who comes to me I will not drive away"'. Instead, Adam and Eve hid and did not confess their guilt. *Psalm 32*, *5* identifies a way forward after wrong-doing and the encouragement to 'acknowledge my sin to thee/and I did not hide my iniquity. I said "I will confess my transgression to the Lord"; then thou didst forgive the guilt of my sin, and a little later in that same psalm: Thou are my hiding place for me' (v. 7). Adam and Eve did not take advantage of this possibility. Luther (1483–1546) writes vehemently on this verse about Adam and Eve's hiding after their Fall seeing it as a deliberate hiding from God Himself: ' . . . they tremble at the sound a shaking leaf ... This fleeing from God is therefore the strongest possible testimony of Adam against himself' (undated, unpaginated). He writes, O! how awful a fall! To fall from the safest security and delight in God into fear and dread so horrible. . . . For it is not the devil from whom Adam and Eve are now fleeing. They are rushing from the sight of God their Creator . . . This dread, therefore, is actually a flight from and a hatred of God himself' (unpaginated).

## 6. Temptation and the 'Fall' of Catholic Education

In this final section, we suggest that over the last 30–40 years, Catholic schools in England and Wales have become in danger of 'falling' to various temptations. By this we mean succumbing to the lurid beckoning of secular philosophies of education which

offer no room for the transcendent, to policies which emphasise that individual worth emerges from examination success and the tempting belief that the securing of well-paid employment will lead to happiness and peace. For those who work in Catholic schools, this is tantamount to a fleeing from that which gives young people the safest security—a sense of the sacred, a delight in God, and a moral participation in a purposeful world to which the whole of humanity may contribute. The 'attractive' and seductive language of 'achievement' and 'success' founded on the notion of competitive individualism is counter to Catholic education which is centred around the pursuit of the common good and the development of a stable self where body, mind, and spirit become developmentally integrated over time. These principles are in danger of erosion if those working in Catholic schools allow themselves to 'fall' victim to distorted understandings of the self and for what purpose humanity was created (Torevell 2019a, 2019b, 2019c, 2020a, 2020b). In a theological sense, it means fleeing and hiding from God, as Luther notes.

Ryan Wilson qualified as an English secondary teacher in 2005; from the age of eight living in Northern Ireland he had dreamed of being a teacher. His autobiographical narrative of his professional life working in Essex and London comprehensive schools captures something of the climate that now exists in schools and colleges, a debilitating 'system' to which Catholic educators like everyone else are likely to fall victim (Wilson 2021). He left the profession at the age of 32 but hopes to return to the classroom one day because he loves teaching so much. He writes that over ten years and by the time he was in senior management, the pressure had ratcheted up from government level and the overriding emphasis on examination results was enormous which coincided with budgets were being cut. One of the reasons for writing the book was because it was important that people became aware of the pressure Ofsted, the inspectors, put on those involved in education. It passed on to the head teachers, because their jobs lived or died by the examination results they got and then the pressure stepped down to the leadership teams, to heads of department, and to teachers, and they inevitably passed it on to the pupils. Everyone became terrified and stressed. The obsession with data and statistics led to fear. Wilson argues fairly that some monitoring of data is useful and important, but not when it became an obsession and raised the stakes so high. That was what Ofsted inaugurated. There was a real sense the Ofsted would look at the data and make a judgement on your school on the basis of examination results before they even arrived. That is where the culture of fear came from and that is why so many talented teachers started to leave the profession as he did (161–166). One of the authors of this article echoes Wilson's claims as a serving Catholic school leader who continues to work in an area of extreme material and cultural deprivation and has seen first-hand the tensions faced by staff who on a daily basis face the difficult challenges and humiliations that fidelity to the mission that Catholic education brings. Numerous have been broken by a system that reduces young people to a number—a grade 9 is outstanding yet a grade 1 is worthless.

As Catholic educators become swamped by governmental initiatives and educational thinking whose underlying philosophy is at odds with that of Catholic philosophy of education, schools and colleges inevitably 'fall' victim to irresistible forces. Unsurprisingly, feelings of guilt and shame arise among Catholic teachers as they capitulate to a 'system' that has very little room for the promotion of creativity and the fostering of the spiritual and moral growth of students and which measures success by the number of students they get to achieve top grades at GCSE and A levels and who move on to Russell Group Universities and high prestige apprenticeships.

The Catholic community established its mission to education to provide schools for the poor. In the years after the emancipation of the Catholic Church in England in 1829, schools were built before parishes. This initiative was a courageous act of love to stand in solidarity with the poor. Writings about the heart of Catholic education claim that its core vocation is to serve the needs of those who are limited in the material goods of the world (Grace 2015). If lost sight of, then Catholic education falls short of the ideal for which it was established. Catholic education seeks to build the Kingdom of God and offer the poorest in

society the best. Such courage embraces a positive anthropology which reflects the Biblical teaching that all human beings are gifted by their common dignity and are made in the image of the Creator, *imago dei*, but also that they are prone to failure.

The feelings of guilt and shame that often come with the public humiliation of teachers at a national level as Inspection Reports place, for example, a school in special measures is frequently at odds with their sense of vocation. A jewel in the crown of Catholic education is that it has always served in areas of economic and cultural deprivation. However, with the shaming of individuals based on an agenda that fails to give credence to so-called 'soft data'—for example, the building of positive person-centred relationships with students, walking with them on their faith journey, educating their whole person, and serving with agapeic love—one has to ask if Catholic education can survive an onslaught which reduces those made in the image of God to economic units.

Demanding external pressures from a swathe of stakeholders inevitably impinge on the ethos, management, and teaching of Catholic schools who are part of a much larger system of governance and to which they owe allegiance. The 'fall' which is likely to occur echoes the narrative of temptation in *Genesis 3* and shows how a lack of faith and courage can result in falling out of love, which is as natural as falling in love. *Genesis 3* is the flip side of *Genesis 1*, in which God says five times that creation is good and illustrates that while human beings are 'good', they are also prone to fall. To fall, to fail, to sin, is to be human.

The Second Vatican Council document *Gravissium Educationis* teaches that ' . . . a true education aims at the formation of the human person in the pursuit of his ultimate end and of the good of the societies of which, as man, he is a member, and in whose obligations, as an adult, he will share. . . . Therefore children and young people must be helped, with the aid of the latest advances in psychology and the arts and science of teaching, to develop harmoniously their physical, moral and intellectual endowments so that they may gradually acquire a mature sense of responsibility in striving endlessly to form their own lives properly and in pursuing true freedom as they surmount the vicissitudes of life with courage and constancy.' (https://www.vatican.va/archive/hist_councils/ii_vatican_council/documents/vat-ii_decl_19651028_gravissimum-educationis_en.html, (accessed on 29 August 2021)).

Catholic educators might seek during this difficult time to go back to earlier thinkers who established the Church's commitment to educating the poor. This might be a *kairos* moment, a time of great opportunity to see the broken state of some Catholic schools as an opportunity to reassess, revaluate, and restore Catholic education to its intended state. A good starting point might be to look at two of the founding figures of Catholic education: St. John Baptise De La Salle and St. John Bosco. De La Salle's *The Conduct of the Christian Schools* reflects a time of 'fall' from the ideals of Catholic education in France when education was sadly the preserve of the rich, a view that was solidly upheld by the Church of the day. De La Salle challenged this and established a mission to take education, the great liberator and enabler, to the masses; the poor and unloved. In doing so, he and his companions faced much opposition, even persecution. Yet, he and his order of 'The Institute of Brothers of Christian Schools' were steadfast in their mission to educate the poor no matter how far this belief was to challenge the paradigm of the day. Their ' . . . mission . . . thrived on providing a human and Christian education to young people, especially the poor', (https://lasallian.info/what-we-do/mission/ (accessed on 28 August 2021)).

St John Bosco, the founder of the Salesian order built upon the work of De La Salle and established a mission to educate the ragged children of Turin. Like De La Salle, Bosco faced much opposition from both civil and ecclesiastical authorities for living the maxim 'blessed are the poor'. Bosco sums up his vision of educate in the following ways: 'Do you want to do a good deed? Teach the young! Do you want to perform a holy act? Teach the young! Do you want to do a holy thing? Teach the young! Truly, now and for the future, among holy things, this is the holiest' (https://www.bosco.link/index.php?document_srl=36910&mid=webzine (accessed on 27 August 2021)).

Let us now outline a little more the 'culture' and *zeitgeist* in which Western European Catholic schools and colleges and universities are currently embedded. It is a climate of fear which educational professionals are tempted to endorse and which reflects a false estimation about what education is about and what it is for. Taylor suggests that a 'post-Durkheimian dispensation' (Taylor 2007, p. 487) has now occurred in the West, by which he means that religious adherents previously 'closely linked to their insertion in their society' are no longer evident (491). There has been a post-war, dangerous slide in our social imaginary which has destabilized and undermined a unified social and religious fabric. The impact of powerfully enforced social differentiation brought about largely through secularism and consumerism now errs dangerously into social approval and opprobrium rather than towards the imperative of love, which has resulted in a destabilizing effect on society. Conflict is provoked, advancing the collapse of a civilized order (Taylor 2007, pp. 488–92).

In a Western culture where religious beliefs and practice have been recalibrated (Davie 2010; Norris and Inglehart 2004; Taylor 2007, 2015, 2017), Taylor uses the word 'secular' to describe how explicitly religious forms of living are now simply one option among many. The impact upon education has been telling. He claims three spaces or categories exist to describe this shift. The 'middle ground' (the second category) is the space or category between those who do not hold a strong religious sense (the first category) and those who hold its opposite—a sense of God's absence—resulting in feelings of alienation, *ennui*, and even nihilism (the third category). Although such spaces are porous like Mellor and Shilling's (2014) four categories to describe the process of secularization, they nevertheless help us to understand what is happening in post-traditional cultures and how they have impacted on educational policy and practice.

Leading philosophers and sociologists of education argue that educators have to guard against falling victim to two main enemies: neo-liberalism and globalisation (Arthur 1995; Bryk et al. 1993; Grace 2015; Pring 2018). These influences are characterised by the overwhelming impact of the market, the justification of educational expenditure on education only when it secures economic benefit, the operation of a consumer choice mechanism, the emphasis on competition, and the drift to commodification, all leading to the conclusion that educational theories are now founded upon a monetary paradigm, a 'human capital theory', which de-humanises the entire project and which is deeply at odds with *Genesis'* teaching on *imago dei.* The imperative to love finds little room in this catalogue of cultural forces. Senior managers and teachers may become (sometimes unknowingly) influenced by policies and practices which are in direct tension with the Christian perspectives witnessed in *Genesis 1* and *3* and slide towards pride, self-autonomy, and consumerism allied to the shameful hiding of such dangers behind a veil of accountability and institutional survival. The 'idols' that start to be 'worshipped' in this situation reflect the Edenic couple's susceptibilities and failings. Christian higher education has not escaped these influences which undermine a confident sharing of a clear Christian vision for education and for society (Docherty 2018; Furlong 2013; Sullivan 2019). As Furlong bluntly puts it, 'Newman is dead' and his vision of the 'cultivation of the mind', which is worth 'seeking for its own sake' (Furlong 2013, p. 7) and which is itself 'a treasure' (Furlong 2013, p. 118), has become extinct. He concludes, 'Overall, a very different world from the financially secure self-governing community of scholars dedicated to the pursuit of truth' (Furlong 2013, p. 116) that used to exist is now evident.

In Bryk et al.'s ground-breaking study of the shift in educational philosophy underpinning school education in America from the 1960s to the 1990s, they argue that Catholic education offers a distinctive alternative to non-faith education, which he refers to as 'public schools' (Bryk et al. 1993, p. 11). His contention is that the latter is 'dominated by market metaphors, radical individualism, and a sense of purpose organised around competition, and the pursuit of individual economic rewards' (Bryk et al. 1993, p. 11). In this, it is not dissimilar to the educational context in the UK. For these authors, what is at the core of Catholic education is an emphasis on love and the common good and the

fostering of human co-operation in pursuit of such goals. Jonathan Sacks (Sacks 2020) in his *Preface* to *Morality: Restoring the Common Good in Divided Times* warns that 'The free market and liberal democratic state together will not save liberty, because liberty can never be built by self-interest alone. I-based societies will eventually die'. Those reductionist features of education centred around individualism and economic success identified by Bryk et al. (1993) have, according to Sacks, been exacerbated over the past 25 years.

The stance taken in the 1930s and 1940s by one of the greatest of all Catholic philosophers, Jacques Maritain (1882–1973), reflects the teaching of *Genesis 3.* He offered a distinctive educational anthropology based on the notion of the person formed largely by moral forces, socially connected to the rest of humanity and who has a spiritual life which can be encouraged and directed towards a greater good, rather than self-aggrandizement (Maritain 1943, 1946). Emphasizing the virtues of love and wisdom, he argued for the common good and claimed that increasing secularization was undermining this pursuit. Schooling should form, by contemplative as well as practical means, a moral and spiritual disposition towards justice, courage, beauty, and truth (Torevell 2010, 2019a; Zajonc 2016). Like Sacks, he offered a countervailing voice to educational thinking which denied there is any such thing as moral truth and which sees paid employment as the primary aim of education (Eagleton 2015).

Grace (2015) has been writing about market-driven philosophies of education versus more holistic ones for some time now. As early as 2002, he suggested that the major challenge for Catholic school leaders resided in this clash: 'If a market culture in education encourages the pursuit of material interests what becomes of a Catholic school's prime commitment to love of one's neighbour? And if the calculation of personal advantage dominates how can Catholic schools remain faithful to values of solidarity and community?' (Grace 2002, p. 199). Such approaches detrimentally affect those schools which are in socially deprived areas and cannot compete with schools in more affluent neighbourhoods, who play the 'market game' more efficiently because they have the resources to do so. That is why Grace advocates an explicit communication of the values Catholic schools stand for and is keen to encourage more research on how they might achieve 'mission integrity' (Grace 2002, pp. 189–204).

Pring (2018), in his defence of publicly funded faith schools, draws a distinction between faith and non-faith schools when he argues that clear thinking about the nature of personhood is one key strategy of survival for Catholic (and faith) schools and colleges. He argues that the religious version of autonomy is very different from the secular one, as it relies on an understanding of the person drawn from sacred Scripture and tradition and includes a form of reasoning which gives pride of place (pace Rahner 1966, 1968) to a sense of 'mystery' to human life and a human ideal to be sought, based on revelation and the person of Christ (Pring 2018, pp. 87–88; Whittle 2016). Human beings are not 'independent' because they depend for their existence on God's creative act of love as Adam and Eve did and are sustained by their intimate relationship with Him. Much discourse about learning and teaching centres around students' independent learning and the encouragement of self-esteem and clearly this can be a legitimate aspiration in light of some young people's debilitating need for dependency and lack of self-confidence. However, as Pring points out, the religious version of autonomy is very different from the secular one since it relies on an understanding of the person drawn from sacred Scripture and tradition which reflects a form of reasoning which places importance on the 'mystery' of life related to an ideal to be sought based on revelation and the person of Christ (Pring 2018, pp. 87–88; Whittle 2016). Human beings are not 'independent' in a religious sense, because they depend for their existence on God's creative act of love and are sustained over time by their intimate relationship with God in Christ. Consequently, it is important to use phrases like 'independent learners' with caution, lest they undermine the distinctive theological anthropology involved in religious notions of the self.

Fukuyama draws attention to this (post)modern move towards establishing a strong sense of self. He suggests that although economists assume that most human beings are

motivated by 'preferences' or 'utilities'—desires for material resources or goods—they forget they are far more galvanized by Plato's notion of *thymos*—the craving for positive estimations of their self-worth and dignity. This can come from within as they develop self-esteem, or from without, as others come to acknowledge their distinctiveness (and their struggles) positively. When they internally absorb positive responses to themselves by others, they feel pride; if they are refused these, they feel anger or shame. The rise of identity politics and the contemporary emphasis on equality resides within this establishment of self-worth. Rousseau foresaw that human beings are apt to compare, contrast, and evaluate themselves with other human beings. In *Les Rêveries du Promeneur Solitaire* (*Reveries of the Solitary Walker*), he confesses how he himself became the victim of other people's evaluation (Rousseau 1987, pp. 55–56), and in his *Discourse on the Origins and the Foundations of Inequality Among Men*, he denounces the shift from legitimate *amour de soi* (love of self) to *amour-propre* (self-love or vanity) which leads to feelings of pride and to the frequent use of contrasting words like strong and weak, swift and slow, and fearful and bold (Rousseau 1997). It is a theme Taylor picks up in his analysis of the bourgeoning of a secular age (Taylor 2007).

## 7. Conclusions

The oppressive environment in which Catholic schools presently operate demands that senior management teams, staff members, and students alike are driven to achieve examination and employment success as their primary aim based on a system of intense managerialism, an obsession with data, endless inspections, and competitive league tables, all of which contribute to producing high levels of stress for all those involved. It also entails the leaving of the profession by droves of teachers on moral and health grounds. Ryan Wilson's story is emblematic of one of thousands. Not surprisingly, mental health issues become rife in such a climate (Torevell 2021), and the temptation to succumb and agree to this destructive system is immensely strong—the fruit is sometimes just too inviting since professional worth and reputation rest upon this pact with the devil. *Genesis 3: 1–18* offers a warning about the consequences of such a 'fall'—feelings of alienation and shame within professionals as they agree to what they know in their hearts through their experience and knowledge, to be a reductionist and damaging understanding and practice of education. Masaccio's painting of *The Expulsion* is a vivid visual reminder for all time of succumbing to the lure of insidious and dangerous attractions and the dire consequences of such capitulation. Nevertheless, the narrative does offer hope towards the end—you do fall, never hide your mistake but rather admit your capitulation and seek forgiveness—a position Adam and Eve were not able to adopt.

**Author Contributions:** Both authors were equally involved with conceptualization, methodology, writing the draft, as well as review and editing. All authors have read and agreed to the published version of the manuscript.

**Funding:** This research received no external funding.

**Conflicts of Interest:** The authors declare no conflict of interest.

## Notes

1    The *Genesis* story has also been used to defend misogynistic claims about the identity of womanhood. The Fall is the woman's fault, not the man's. Over history, the narrative has been interpreted to indicate how women are inherently seductive, innately temptresses, disgracefully bodily and generally not to be trusted. Not surprisingly in response, feminist critiques in the 20th and 21st centuries have endeavoured to correct this misguided interpretation.

2    The work of Girard and his notion of mimetic rivalry extends some of the implications of the Genesis text (Girard 2005).

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
