# Peer review of "The Naked Truth: Temptation and the Likely ‘Fall’ of Catholic Education"

_religions, doi:10.3390/rel12110958_

Round 1
Reviewer 1 Report
This is an interesting and engaging article. The use of the painting is an interesting approach and there is great merit in the use of Genesis 1 and 3. I think it still needs some work, mostly around clarification.
- The translation of Adam as ‘ground’ or ‘earth’ is very good, but as they can mean two different things, I think the authors need to choose one or the other. Although there is merit in approaching the term in both senses of the word.
- The first section 1. A Universal Story of Envy, Pride…. Could do with a more systematic approach. It is very comprehensive, but it seems to jump around from source to source.
- Line 129 – I think the inclusion of Nazism is not quite Godwin’s Law, but the use of Nazism and the sophistication of the arguments leave more questions unanswered than answered, and I think if your use Trump and Johnson (which I would question), you need to sue their names or at least their titles, considering you did when it came to President Biden. Also, if the authors discuss the fall in the political terms, does it advance the argument of the article, considering the authors don’t return to it when discussing Catholic Education.
- Line 185 – I think that the Masaccio’s piece could do with more referencing from art historians and more academic sourcing in terms of the explanation of the painting.
- Line 218 – should the reference to St. Francis de Sales’ work be added.
- Line 222 – I though the ‘sword’ trope discussed here was very interesting and might be worth expanding.
- Line 293 – The understanding of manliness and the 15th century could also merit a reference.
- Line 305 – The referencing to Milton. I think that it might merit a section of its own. It is unusual that in the section on Augustine, it goes straight into Milton. The authors might also be interested in exploring Book Nine of Paradise Lost, beginning around line 885, and the quandary that Adam finds himself in – if he eats the apple he looses paradise, and if he doesn’t, he loses Eve.
- Line 467, if this is the first reference to Wilson, perhaps a little more here.
- Line 488, the authors might explain the humiliations and the guilt and shame referenced. Anyone in the area of Catholic Education might agree with the authors, but it needs better explanation.
- Line 512. The biggest issue is that the term ‘fall’ is not sufficiently explained within the context of Catholic Education and it needs to be.
- Line 560. Should the authors focus on one area of Catholic Education – schools or colleges or universities. Also, should it be more focussed to one country – Britain perhaps. Many of the issues that arise are distinctive to Catholic education in mostly the Western world.
- Line 566. Post-war – does it need to be specific. Also, should we move on to another post- period considering the nature of the change in education even in the last 30-40 years.
- This section is heavy on referencing (which is excellent), but it makes the lack of sourcing in previous section look unbalanced.
- It is interesting that there is no mention of parents as a source of pressure on teachers in a similar manner to OFSTEAD
- In terms of the sourcing – there needs to be more of an integration of sources. There is a touch of first this sources, then this, then this next one, etc. In places the sourcing is very well integrated; in other places not.
- The Conclusion doesn’t have any reference to the painting, and I don’t think that all the loose ends of this excellent piece of research are tied up.
Author Response
All comments have been addressed. The political dimension of the article has been taken out as suggested. Arts historian voices have been added
and the first section has become much more focussed and systematic. The ‘sword’ trope has been expanded upon and the issue of teachers’ guilt
and shame addressed. Wilson has been introduced more carefully and the ‘fall’ been much more fully explained. The Integration of sources has
been improved as suggested. The focus is now exclusively on Catholic schools and the Conclusion refers to the fresco again as it does at the start.
de Sales ha been referenced and post-war taken out and last 30-40 years added. The issue of manliness in the 15th century has been explained
more fully. ‘Earth’ has been chosen and ‘ground’ deleted. The reference to Milton has been added to.

Reviewer 2 Report
The idea and the problem of the research are actual and interesting from different viewpoints, but the article seems not finished. In the beginning, there were announced that it will be 5 aspects of analysis, but only 4 are presented in the article. As well as conclusions. There are no conclusions at all. Only the place for them is occupied for it in the article.
After reading the full version: The topic of the article is interesting and quite actual, and the article itself enables us to look at the main question from different sides and to create a better understanding of it. Division to 5 parts presented at the beginning of the article enables to follow the main research question and to analyze it from different aspects. All main features of the scientific articles and the research are kept, conclusions are based on the theoretical and empirical discourse analysis.
Author Response
References have been checked and the entire article revised carefully.
